# What Is the Psychological Role of the Virtual Self in Online Worlds? A Scoping Review

Adrià Gabarnet [1], Guillem Feixas [1,*] and Adrián Montesano [2]

1 Department of Clinical Psychology and Psychobiology, Institute of Neurosciences, Universitat de Barcelona, 08007 Barcelona, Spain; agabarnet@ub.edu
2 Department of Psychology and Educational Sciences Studies, Open University of Catalonia, 08035 Barcelona, Spain; amontesano@uoc.edu
* Correspondence: gfeixas@ub.edu

**Abstract:** Several studies have researched how people can use the anonymity of the Internet to explore different aspects of their identity. There are many different platforms where one can create a "virtual self" by actively choosing how one presents themselves to others, and each one is designed with different characteristics depending on their purpose: from socialization to professional networking or even entertainment. Different studies have usually focused on specific platforms, but there is no one comparing how people construe their virtual self across different online environments. In this review, we aimed to synthesize research studies carried out about the construal of one's identity within online platforms and how it can relate to different aspects of participants' offline identity, such as their self-esteem or self-concept clarity. Data were gathered from 34 publications that researched this topic across different kinds of online platforms. We conducted a quality assessment and a narrative synthesis, reporting and comparing the main findings, as well as identifying possible gaps in the literature. Many studies have explored the idea of people creating either an idealized or accurate version of themselves when construing their online identities. Others have also contemplated the possibility of exploring identities that diverge from both the actual and the ideal self or even an actively negative one. The latter was studied more in the context of video games and other avatar-mediated worlds. We found that people with low self-esteem create a more idealized virtual identity rather than a virtual self that is similar to their offline self. Other variables such as the purpose for using the online platform or self-concept clarity also had a role in virtual self construal, although the latter was only studied in the context of social media.

**Keywords:** virtual self; virtual environment; online video game; social media; virtual identity

## 1. Introduction

With the spread of virtual communities, human interaction is extending more into online environments, including online video games, social media, dating websites, and other virtual platforms where users can interact with each other such as Second Life or the emerging Metaverse. Some of them have primary functions such as entertainment in the case of online video games or job-seeking, i.e., LinkedIn, and the social aspects are a secondary perk, whereas others, like Facebook[TM] and other similar social media platforms, are designed specifically for socialization, but most of them offer some way to create a virtual identity within the platform. This function can be as simple as a profile picture or as complex as a fully customizable, three-dimensional avatar. Given the high anonymity that most online environments possess, users can decide how to create their virtual selves. They may construe their virtual identity as they are offline, but they could also take the opportunity to pose as an enhanced version of themselves [1–3]. In certain environments, people can even choose to explore alternative identities, a phenomenon that Taylor (2002) [4] named "identity tourism", and is described as the inherent satisfaction

that some people may experience when taking on a different identity for a time. These alternative identities do not need to compensate for any defects or shortcomings of the offline self. Instead, it is the mere act of temporarily "being someone different" that can be intrinsically rewarding.

These new forms of social interaction have given rise to numerous research studies about the virtual self which can be understood as the aspects of a person's identity that they express online. Previous researchers, such as Gabarnet et al. (2022) and Triberti et al. (2022) [5,6], have aimed to ascertain how many different types of virtual self construal people can have, whether it is an ideal version of themselves, the exploration of multiple alternative identities, or even the expression of negative traits they are not comfortable with in the offline context, as observed by Hu et al. (2017) and Mancini and Sibilla (2017) [7,8]. However, to our knowledge, no study has taken the task to gather and synthesize the different proposals, and to explore how they may intersect.

Some researchers have explored how different variables can lead to different forms of virtual self construal. Studies like the ones conducted by Dengah and Snodgrass (2020), Forest and Wood (2012), and Michikyan et al. (2015) [1,3,9] explore the relationship between self-esteem and the way people create their virtual self, but there have also been researches like Fullwood et al. (2016, 2020) and Strimbu and O'Connell (2019) [10–12], who have explored other personal variables such as self-concept, or Behm-Morawitz (2013), Peng (2020), and Ranzini and Lutz (2017) [13–15], who studied the motivations people have for using online platforms. On the other hand, studies like the ones by Leménager et al. (2013) and Looy et al. (2014) [16,17] have also investigated how the type of virtual self construal could be related to the users' psychological well-being or the excessive usage of the online platform.

For several decades now, the importance of self-identity has been recognized as an essential factor for psychological maturity and emotional well-being (e.g., Pilarska (2014) [18]). This vast construct of self-identity encompasses many different psychological functions and processes such as self-clarity, understood as the extent to which an individual's beliefs about themselves are clearly understood and consistent over time [19] or self-presentation, which is defined as how an individual transmits information about themselves to others, either knowingly or unknowingly [20].

Many aspects of traditional self-identity, like self-esteem or self-presentation, could be applied specifically to the aspects of an individual's identity that are construed within online virtual environments such as video games or social media. Research on the topic of virtual identity is of great relevance nowadays, since the use of online environments is present in the daily life of increasingly more people and can influence their offline behavior and quality of life to some extent [21,22]. Furthermore, there has been an increasing concern in recent years about what may constitute an abusive use of virtual environments. Even with the inclusion of internet gaming disorder in the DSM-5 [23], the exact boundaries between useful, recreational, or problematic usage of these platforms is still unclear. Understanding the current state of the art regarding how people construe their virtual identity and its potential link with offline identity can be useful for anyone who intends to research this emerging topic.

The aim of this review is to explore the current literature about:

(1) The different ways in which people construe their virtual identity in online environments.
(2) How this relates to their offline identity.
(3) The psychological function that this virtual self might play in their lives, whether it serves as a compensation for perceived offline deficiencies or as a way to explore alternative identities altogether.

We also intend to identify the different kinds of research conducted on this topic, as well as any possible gap that may exist in the current literature. Finally, the study also aimed to synthesize the different findings that have been made by previous research on virtual identity and propose future lines of research on the topic. Given the broad scope

of these objectives, oriented to provide a general idea of the current state of the art of the literature about virtual identity construal, a scoping review was considered to be the most suitable methodology for this study, as this methodology can be useful when the amount of available publications or the broadness of the research topic are not suited for more precise methods such as systematic reviews [24].

Previous literature reviews have been published regarding specific effects of video games, such as the one by Chan et al. (2022) [25], or social media use, like the one by Almutairi et al. (2022) [26]. One review by Sibilla and Mancini (2018) [27] even studied the relationship between online environment users and their avatars. However, no reviews have been conducted that tackle this issue in social media. There are also no reviews that compare the results found in different types of online environments. Therefore, we decided to conduct this review including any type of virtual online environment since this would allow a comparison between different platforms.

## 2. Methods

### 2.1. Target Population

The population of interest for the articles included in this review were users of online virtual environments. There were no age restrictions, although articles that focused on specific age ranges were considered when charting and reporting the data.

### 2.2. Search Terms and Strategies

The review was conducted following the PRISMA-ScR indications [28] and was pre-registered at OSF Registries (DOI removed for anonymity reasons). We developed a search strategy to gather all available knowledge about virtual self construal and its relation to personal identity. The formula aimed to identify all articles that mentioned four principal concepts in their title, abstract, or keywords: terms related to online environments, personal identity, online identity exploration, and virtual representations. The following strategy was developed, following the PRESS 2015 checklist [29]: Virtual environments AND Self-concept AND Idealized self AND Avatar. For each of the search terms included in the formula, a list of alternative possible terms was also included.

Alternative terms related to virtual environments were "virtual communities", "internet", "websites", "videogames", "social media", "online environments", "virtual platforms", "virtual reality", "social networks", and "augmented reality".

Alternative terms for self-concept included "self-esteem", "self-worth", "self-evaluation", "self-thought", "self-confidence", "self-definition", "self-image", "self-perception", "self-construction", "self-construal", "identity", "personal identity", "self-knowledge", "self-hood", "self-referent", "self-reference", "self-presentation", and "self-representation".

Alternative terms related to identity exploration online were "Idealized Self" included "ideal self", "ideal version", "projection", "project", "identity tourism", "virtual tourism", "actual self", and "self-enhancement".

Alternative terms related to avatars and other forms of virtual representations included "character", "profile", "virtual self", "virtual identity", and "virtual representation".

With the search algorithm resulting from this, over the months of October and November of 2021, we searched in the databases SCOPUS, PubMed, PsychInfo. and the Cochrane Library for articles published within the last 10 years. The BioMed Central ISRCTN Registry database was also consulted to account for unpublished studies and grey literature. Additionally, a hand search was performed on four key journals (*Computers in Human Behavior*, *Games and Culture*, *Games for Health*, and *Cyberpsychology, Behavior, and Social Networking*) to look for relevant articles. A total of 212 articles were identified by the search engines, and 99 more were found with the hand-searching strategy.

### 2.3. Study Selection Process and Criteria

The following criteria were applied to filter the articles and keep only the ones that were relevant to this review. The publications needed to: (a) be empirical research studies,

(b) be written either in Spanish or English, (c) have been published within the last 10 years (2011–2020, both included), (d) have the full text available, (e) be focused on users of an online virtual environment (i.e., social media or online video games), (f) have a measure of personal identity in any of its expressions (e.g., self-esteem and self-concept), and (g) explore the participants' online identity, either as its main focus or as a secondary objective.

We defined online virtual environments as any digital platform where users can interact with each other and present themselves to others through a customizable profile or avatar.

The filtering process was performed by three researchers working independently: two psychology undergraduates and a predoctoral researcher. They individually read the abstract of all the articles to check whether they met the inclusion criteria. Regular coordination meetings were held between the three researchers to compare their assessments. To resolve discrepancies, the full text of the article was read and discussed to reach a final decision. When the full text was unavailable, the authors were contacted asking them to send it. If we were unable to obtain the full text of the article by any means, it was excluded from the review.

The process took six months (November 2021—April 2022) during which a snowball search was performed, inspecting the references of relevant articles to find any publication that could have been missed. By the end of this process, 153 more articles were found, and after applying the inclusion criteria, 34 were included in the review (Figure 1). The kappa index computed between the three raters for the filtering process (kappa = 0.48, *p*-value < 0.001) indicated an intermediate to good inter-rater agreement, according to the criteria by Fleiss et al. (2003) [30].

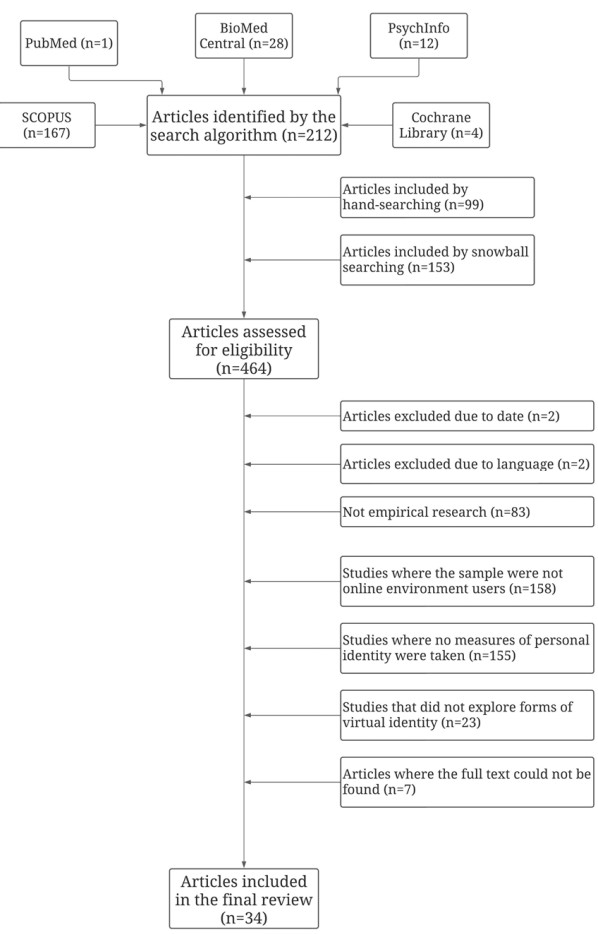

**Figure 1.** Flowchart of the search and filtering process with the number of articles that were included or excluded in each stage.

*2.4. Quality Assessment*

We also performed a quality assessment to evaluate the included articles. The aim was not to exclude poor-quality articles but to consider the different qualities of the articles when reporting their findings. A rubric was designed based on the STROBE criteria [31] and edited to account for how relevant the article was to the review. Note that the rubric was not meant to assess the quality of the study designs but the reports. The rubric assigned each article a score between 0 and 40 and classified them in one of four ranks: D (0–10), C (11–20), B (21–30), and A (31–40).

The quality assessment was conducted independently by the same researchers. After a brief 2 h training period with the rubric, each researcher independently read the full text and rated every article using the rubric. As with the study selection, regular meetings were held to maximize consensus. Any discrepancies in the rank of an article were discussed and the full text was read carefully to reach a final decision. Despite this, the inter-rater agreement for the quality assessment (kappa = 0.34, *p*-value = 0.1) ended up being poor according to the criteria by Fleiss et al. (2003) [30].

*2.5. Data Charting*

After all articles were rated, they were charted by the same three researchers in a database that gathered the most important data: the article's title, authors, year of publication, county, quality rank, study design, type of intervention, duration of intervention, instruments used, sampling method, sample characteristics, virtual environment studied, main objectives, theoretical framework, statistical analyses used, predictor variables, outcome variables, modulator/interaction variables, and main findings. The process was once again completed individually by each researcher and the databases were compared, to optimize the final database. Finally, a narrative synthesis of the charted data was conducted solely by the predoctoral researcher who also took part in the previous stages, reporting it in a comprehensive and structured way.

**3. Results**

Of the 34 reviewed articles, 15 (44.11%) focused on social media websites, seven (20.59%) researched online video games, seven more (20.59%) focused on avatar-mediated online environments, such as Second Life, and three (8.82%) studied specialized online platforms, such as dating apps. Additionally, there were two studies (5.88%) that included more than one type of online environment. Cacioli and Mussap (2014) [32] included both video games and avatar-mediated online environments, whereas Suh (2013) [33] used the more general term of "online communities", which included both social media and avatar-mediated environments. The most prevalent measures of personal identity used in the included studies were self-esteem, self-efficacy, and self-concept clarity. On the other hand, virtual identity construal was mostly explored by measuring the discrepancy between the participants' self-presentation in virtual environments and their own assessment of their offline self. Some studies complemented this by comparing the participant's virtual identity with other aspects of themselves such as their ideal self [1,34] or their "negative true self", understood as those traits that an individual recognizes as part of their identity but does not express due to social constraints [7]. In some studies, the discrepancy between a person's virtual identity and their actual self was conceptualized as "deceptive behavior" [14,35]. Additionally, many studies complemented their research with other variables related to the participants' psychological well-being, such as anxiety [32], narcissism, or self-objectification [36], or related to their online behavior, such as pathological gaming [17], their main purpose for using a specific online platform [14,21], or whether they felt safer expressing themselves online rather than offline [9,10]. Table 1 shows a summary of the included articles.

**Table 1.** Summary of the articles included in the review, divided by the type of virtual environment they studied.

| Author and Year | Country | Journal | QA Rank | Study Design | Theoretical Framework | Sample |
|---|---|---|---|---|---|---|
| **Online Video Games (7 Articles)** | | | | | | |
| Carrasco et al. (2018) | Australia | *Proceedings of the 2018 Designing Interactive Systems Conference* | C | Exploratory Qualitative Study | Undisclosed | 10 Australian and Ecuadorian older online video game players (65–95 years old; 5 females) |
| Dengah and Snodgrass (2020) | USA | *Games for Health Journal* | B | Mixed Methods Design | Self-Discrepancy Theory | 21 online video game players, mostly college students (Average age of 23; 4 females) 57 online video game players (Average age of 24; 17 females and 1 non-binary person) |
| Ko and Park (2020) | South Korea | *Internet Research* | B | Cross-Sectional Observational | Self-Discrepancy Theory | 347 USA online and mobile video game players (20–80 years old, 185 females) |
| Leménager et al. (2013) | Germany | *European Addiction Research* | B | Case–Control Study | Self-Discrepancy Theory | 45 online video game players, addicted and non-addicted, as well as non-players (average age of 26.33; 15 females) |
| Looy et al. (2014) | Belgium | *Multiplayer: the social aspects of digital gaming* | C | Cross-Sectional Observational | Self-Discrepancy Theory | 304 European World of Warcraft players (average age of 24.54; 49 females) |
| Mancini and Sibilla (2017) | Italy | *Computers in Human Behavior* | B | Cross-Sectional Observational | Self-Discrepancy Theory | 854 online video game players (14–62 years old, 236 females) |
| Przybylski et al. (2011) | UK Germany USA | *Psychological Science* | C | Cross-Sectional Observational | Self-Determination Theory | **Study 1**: 144 college students (Average age of 19.83; 96 females) **Study 2**: 979 online video game players (18–48 years old; 150 females) |
| *Avatar-Mediated Environments (7 Articles)* | | | | | | |
| Behm-Morawitz (2013) | USA | *Computers in Human Behavior* | B | Cross-Sectional Observational | Social Cognitive Theory | 279 Second Life users, mostly from the USA (18–70 years old; 157 females and 8 male-to-female transgender people) |
| Gilbert et al. (2014) | USA | *Computers in Human Behavior* | B | Exploratory Qualitative Study | Undisclosed | 24 Second Life users (18–64 years old, 18 females) |

**Table 1.** *Cont.*

| Author and Year | Country | Journal | QA Rank | Study Design | Theoretical Framework | Sample |
|---|---|---|---|---|---|---|
| Hooi and Cho (2013) | Singapore | *Computers in Human Behavior* | A | Cross-Sectional Observational | Self-Categorization Theory and Self-Perception Theory | 159 Second Life users (older than 18; undisclosed gender) |
| Kim et al. (2012) | South Korea | *Computers in Human Behavior* | B | Cross-Sectional Observational | Social Cognitive Theory and Self-Categorization Theory | 111 Second Life users (undisclosed gender and age) |
| Koles and Nagy (2012) | Austria | *Journal of Virtual Worlds Research* | B | Exploratory Quantitative Study | Self-Congruence Theory | 153 Second Life users (18–73 years old; undisclosed gender) |
| Messinger et al. (2019) | Canada USA | *Journal of Business Research* | B | **Study 1**: Mixed Methods Design **Study 2**: Case—Control Study | Self-Enhancement Theory and Self-Verification Theory | **Study 1**: 167 USA college students (average age of 20.23; 94 females) and 97 Second Life users (average age of 30.5; undisclosed gender) **Study 2**: 23 USA college students (undisclosed age; 12 females) |
| Triberti et al. (2017) | Italy | *Annual Review of CyberTherapy and Telemedicine* | C | Case–Control Study | Self-Perception Theory | 87 Italian college students (18–36 years old; 45 females) |
| *Social Media (15 Articles)* | | | | | | |
| Aricak et al. (2015) | Turkey USA | *Computers in Human Behavior* | B | Cross-Sectional Observational | Self-Concept Theory | 459 Turkish college students (18–38 years old; 358 females) |
| Castro and Marquez (2016) | Colombia | *Qualitative Market Research* | C | Exploratory Qualitative Study | Self-Congruence Theory | 15 Colombian college students who were Facebook users (20–25 years old; 9 females) |
| Forest and Wood (2012) | Canada | *Psychological Science* | C | **Study 1**: Cross-Sectional Observational **Study 2**: Mixed Methods Design **Study 3**: Mixed Methods Design | Social Penetration Theory | College students who were Facebook users **Study 1**: 80 students with an average age of 21.35; 58 females and 5 undisclosed **Study 2**: 177 students with an average age of 19.95; 117 females **Study 3**: 98 students with an average age of 21.18; 77 females |

**Table 1.** *Cont.*

| Author and Year | Country | Journal | QA Rank | Study Design | Theoretical Framework | Sample |
|---|---|---|---|---|---|---|
| Fox and Rooney (2015) | USA | *Personality and Individual Differences* | B | Cross-Sectional Observational | Self-Objectification Theory and Life History Theory | 800 men from the USA who used social media (18–40 years old; all male) |
| Fullwood et al. (2016) | UK | *Cyberpsychology, Behavior, and Social Networking* | B | Cross-Sectional Observational | Self-Concept Theory | 148 UK adolescent Facebook users (13–18 years old, 88 females) |
| Fullwood et al. (2020) | UK Australia USA | *Cyberpsychology, Behavior, and Social Networking* | B | Cross-Sectional Observational | Self-Concept Theory | 405 Adult social media users (18–72 years old; 340 females and 2 transgender people) |
| Gonzales and Hancock (2011) | USA | *Cyberpsychology, Behavior, and Social Networking* | B | Case–Control Study | Objective Self-Awareness Theory | 63 USA college students (47 females; undisclosed age) |
| Hu et al. (2017) | Malaysia | *PLoS ONE* | B | Exploratory Qualitative Study | Self-Determination Theory | 57 Chinese social media users (between 20 and +40 years old; 30 females) |
| Hu et al. (2020) | Malaysia | *Behaviour and Information Technology* | B | Cross-Sectional Observational | Self-Discrepancy Theory | 837 Chinese QQ users (20–60 years old; 438 females) |
| Jang et al. (2018) | USA | *Computers in Human Behavior* | B | Randomized Control Trial | Self-Determination Theory | 132 USA Facebook users (19–74 years old; 73 females) |
| Khan et al. (2016) | Canada | *Computers in Human Behavior* | C | Exploratory Quantitative Study | Social Compensatory Theory and Social Capital Theory | 733 Canadian adolescents who use social media (10–18 years old; 451 females) |
| Longo and Saxena (2020) | Ireland | *Atas da Conferencia da Associacao Portuguesa de Sistemas de Informaçao* | C | Exploratory Qualitative Study | Self-Concept Theory | 10 millennial European Instagram users (23–26 years old; 5 females) |
| Michikyan et al. (2014) | USA | *Emerging Adulthood* | B | Cross-Sectional Observational | Self-Discrepancy Theory | 261 USA emerging adults (average age of 22; 195 females) |
| Strimbu and O'Connell (2019) | Ireland | *Cyberpsychology, Behavior, and Social Networking* | A | Cross-Sectional Observational | Self-Concept Theory | 230 adult social media users (18–35 years old; 175 females and 7 undisclosed) |
| Wallace et al. (2020) | Ireland Spain UK | *Journal of Business Ethics* | B | Cross-Sectional Observational | Self-Concept Theory | 234 Facebook users who promote charities online (average age of 22.98; 166 females) |

**Table 1.** *Cont.*

| Author and Year | Country | Journal | QA Rank | Study Design | Theoretical Framework | Sample |
|---|---|---|---|---|---|---|
| *Specialized Online Environments (3 Articles)* | | | | | | |
| Peng (2020) | China | *Internet Research* | B | Mixed Methods Design | Interpersonal Deception Theory and Impression Management Theory | 309 users of a Chinese dating website (18–45 years old; 160 females) |
| Ranzini and Lutz (2016) | The Netherlands Norway | *Mobile Media & Communication* | B | Cross-Sectional Observational | Self-Presentation Theory | 497 USA Tinder (dating app) users (average age of 30.9; 218 females and 1 "other") |
| Sievers et al. (2015) | Germany | *Computers in Human Behavior* | B | Mixed Methods Design | Impression Management Theory | 63 German XING (professional network) users (average age of 26.98; 7 females) |
| *Mixed Environments (2 Articles)* | | | | | | |
| Cacioli and Mussap (2014) | Australia | *Body Image* | B | Cross-Sectional Observational | Self-Discrepancy Theory and Self-Objectification Theory | **Study 1:** 133 avatar users (18–62 years old; all male) **Study 2:** 131 Internet users (18–68 years old; all male) |
| Suh (2013) | South Korea | *Computers in Human Behavior* | B | Cross-Sectional Observational | Self-Discrepancy Theory, Social Cognitive Theory, and Self-Concept Theory | 299 "Virtual Community" (social media or avatar-mediated environments) users (between 20 and +50 years old; 141 females) |

### 3.1. Narrative Synthesis

To report the findings of the studies according to their relevant contexts, the studies were divided into four groups depending on the type of virtual environment they explored.

#### 3.1.1. Online Video Games
Main Findings

While all online games offered some possibility for virtual self construal, MMORPGs were the genre that granted the most options for creating a virtual self, due to their character customization mechanics and avatar-to-avatar interactions. Przybylski et al. (2012) [34] found that when players experienced a high discrepancy between their virtual and their actual (offline) selves, having a character that resembled their ideal self more was related to a higher intrinsic motivation for playing. When players were more satisfied with themselves (the discrepancy between actual and ideal self was lower), characters that resembled their ideal self did not affect their intrinsic motivation. On the other hand, when the character was perceived as more similar to their actual self, players reported increased satisfaction when playing with it. Nevertheless, despite being heavily tied to satisfaction with one's self, this study did not include a specific measure for self-esteem, so we cannot be sure if including it as a variable could affect the results.

This tendency to create idealized virtual identities was observed by most of the studies. Cacioli and Mussap (2014) [32] reported that men tended to create characters that were more attractive than them, especially when concerns about their offline appearance were higher. But this idealized construal of the virtual self was not limited to physical attributes. Dengah and Snodgrass (2020) [1] observed that players tended to create characters that presented less of their negative personality traits than their actual selves although not necessarily more positive ones. That was not the case for players with low self-esteem, who did create characters with more positive traits than their actual self. Almost the opposite happened with high self-esteem players, who created characters that possessed the same amount of negative traits as their actual self, but less positive ones [1]. This seems to indicate that, although there was a general tendency to enhance themselves through their characters, this behavior was closely linked to the player's low self-esteem. Another study by Leménager et al. (2013) [16] observed that addicted players scored lower on their offline body image, self-esteem, and self-concept than non-addicted players or non-players. Addicted players were more likely to create idealized characters than non-addicted players. Despite this, pathological gaming is not motivated by a perceived discrepancy with the ideal self, although idealized characters may be related to player enjoyment [17]. In their study regarding virtual identity and pathological gaming, Mancini and Sibilla (2017) [8] observed that online video game players rated their virtual self as similar to their ideal self in all the Big Five scales [37], except for openness, and no difference was found between pathological and non-pathological players, except for the neuroticism scale, where addicted players would present more idealized characters than non-addicted players.

However, idealization was not the only type of virtual identity construal. In their study, Mancini and Sibilla (2017) [8] observed four different types of characters that players would create.

The character as an extension of the self, which in accordance with the previously mentioned studies was mostly idealized across all Big Five scales.

The character as similar to the self, which mostly resembles the player.

The character as other than the self, which was different from the actual self, although not necessarily more positive, since it was distant from the ideal self in some traits.

And lastly, the character as an antithesis of the self, which was not only different but also considerably more negative than the actual self. Moreover, this last type of virtual identity was observed to be the most common among the study participants.

This opens the possibility of video game characters not being limited to be either an idealized or an accurate version of the player's offline self, but offering more options for exploration of alternative identities, some of which may even be seen as negative by the

player, with them being able to be "bad" in a safe environment without physical or legal consequences.

### 3.1.2. Avatar-Mediated Environments

Given the nature of certain virtual environments that contain elements of both video games and social media without belonging entirely to either of them, these were separated into their own category named "Avatar-Mediated Environments". These environments are characterized by being open worlds where users can interact with each other through customizable avatars but do not have any particular objective aside from socialization. The popular software Second Life was included in this category and encompasses most of the articles included in it.

### Main Findings

Exploratory studies found that avatars could fulfill different functions for their users. The way users create their avatars differs depending on their desired usage in the virtual environment. Gilbert et al. (2014) [21] observed that different functions were assigned to the users' main avatars (the ones they used the most) and to their "alts" (alternative personas that users spend less time with and that usually fulfill very specific functions). According to Gilbert et al. (2014) [21], the prime avatar was mostly used to maintain a consistent identity within the virtual environment. Users poured many traits from their offline self into this identity, but some of them were enhanced. Alts were used mainly to anonymize their primary avatar in environments where it could be recognized but also when they wanted to explore alternative behaviors or practices that would not be consistent with the identity they had created for their primary avatar.

Another psychological function that was shared between primary avatars and alts was what Gilbert et al. (2014) [21] called "Reverse-Enhancement", which referred to the improvement of offline skills or traits after having experimented with them in the virtual environment. This reverse-enhancement could even occur in people who presented pathological social impairments who became more extroverted and improved their social skills after having engaged in the virtual environment for a long time. This effect is actually one of the main components of therapies that involve virtual reality, since they provide an immersive but safe environment where patients can develop changes that can later transfer to their daily lives [38,39].

The main motivations for being in the virtual environment could also affect how the avatar is created and how it influences its user. Triberti et al. (2017) [6] reported that avatars created for leisure were seen by the user as similar to their actual self. When the avatar was created for job-related purposes, female users perceived them as more different from their actual self, while male users showed no difference between leisure and job-related avatars. However, it must be noted that this study was the one that obtained the C rank in the quality assessment. When analyzing how users created their avatars in virtual environments, different studies observed that avatars tended to resemble the users' actual self, although some aspects were enhanced, especially aesthetical traits. The more negatively users saw themselves offline, the more they enhanced their avatar, similar to the findings from studies focused on video games. Even so, when comparing data gathered from the users themselves and from observers, users tended to perceive their avatars as more similar to themselves than external people, who saw the avatars as more enhanced than their users did.

Behm-Morawitz (2013) [13] reported that people who spent time in virtual environments experienced an increase in their health and well-being. However, this effect was decreased in the case of people who used the virtual environment as a means to escape their reality, whereas people who were motivated by online socialization would benefit more from it. Avatar customization also moderated this effect; those who perceived their avatars as similar to how they would like to be in terms of attractiveness were more likely to report their avatar as being influential on their offline health and well-being. Hooi and

Cho (2013) [35] also observed that when a user's avatar was seen as similar to them in terms of values and personality, they presented higher self-awareness and reported being more attracted to their avatar. On the other hand, when the avatar was perceived as being more aesthetically similar to them, the users were less deceptive in their interactions within the virtual environment. Messinger et al. (2019) [40] also reported that users who created an avatar they perceived as attractive tended to behave in a more extroverted, loud, and risk-taking manner within the virtual environment and that these traits could eventually carry over to their offline self over time.

3.1.3. Social Media
Main Findings

Most studies agreed that although people tend to construe their virtual self on social media as similar to their actual self, they always had a component of self-enhancement (i.e., Castro and Marquez (2017), Forest and Wood (2012), Fullwood et al. (2020), and Strimbu and O'Connell (2019) [9,11,12,41]. Several identity-related traits were associated with people's virtual identity on social media. People whose virtual self was more similar to their ideal self tended to have lower self-esteem and psychosocial well-being. Narcissism and self-objectification were also linked to an enhanced or idealized virtual self on social media [36]. On the other hand, higher self-esteem and psychosocial well-being were associated with less deceptive behavior on social media and less self-enhancement.

Another identity-related trait that was explored is self-concept clarity. It was observed to be related to virtual self construal on social media and was studied in both adult and adolescent populations. In adults, less self-concept clarity was associated with more exploration of multiple identities on social media, and higher expression of their ideal self through their virtual identity. However, Strimbu and O'Connell (2019) [12] reported that higher self-clarity did not predict a closer distance between actual and virtual self in adults. On the other hand, Fullwood et al. (2016) [10] observed that, in adolescents, higher self-concept clarity did predict a closer distance between actual and virtual self on social media. The authors attributed the difference with adults to the fact that adolescents were more used to social media since it had been around for all their lives, whereas adults had to adapt to it. Singularly, people exploring different identities on social media did not always construe an idealized virtual identity. Other studies have explored the expression of the negative true self, traits that people consider as part of themselves while also recognizing that they are frowned upon by society. Hu et al. (2020) [42] observed that expressing the "ought self" (the aspects of the self expected by society) on social media was linked to a lower sense of autonomy and self-acceptance by the user. On the other hand, being able to express their negative true self online raised the users' offline self-acceptance and autonomy.

A study by Hu et al. (2017) [7] reported three types of personality trait descriptors: unequivocally positive, controversial, and unequivocally negative. Despite some participants possessing some of the controversial or even unequivocally negative descriptors as part of their identity, they rarely expressed the negative ones offline due to social norms, which according to the authors could reduce their sense of self-congruence. However, the authors observed that people could achieve self-congruence in virtual environments by embodying their "negative true self" due to the lesser social expectations and the temporal suspension of moral judgement that can occur on social media. Additionally, the connectivity that social media promotes allows users to find like-minded people to interact with, which can have both positive consequences, since people can always find a place to feel accepted, but also negative ones, since radical or dangerous ideals could be reinforced in those spaces.

However, even within a single social media platform, there are several features that offer different ways to express oneself. Because of this, a single person can express a different aspect of themselves in each one of those features. Castro and Marquez (2017) [41] observed that Facebook offers a wide assortment of different features (likes, status updates, profile editing, albums, posts, etc.) which users use with different intentions and in which

they may express different aspects of their identity. They found that people tended to express more of their actual self through their liking behavior and in the descriptions provided in their profiles. On the other hand, people tended to enhance themselves and show a more idealized image through their status updates, profile pictures, photo albums, and posts.

Another observation that Castro and Marquez (2017) [41] reported was that people were not always aware of which part of their identity they expressed through their virtual self. There were situations where people reported portraying themselves in an accurate way, but after assessing their behavior on social media, the authors found that they would actually enhance themselves to some degree.

3.1.4. Specialized Online Environments

We referred as "specialized virtual environments" to those applications or websites designed for specific purposes, such as dating or looking for jobs. We assigned them a separate section in this review due to their specialized nature, since the purpose and functions that the virtual self has for their users can considerably affect how they present themselves.

Main Findings

When researching about online dating, Peng (2020) [14] reported that 83% of users lied to some extent in their profiles, although the overall severity of deception was not high. On the other hand, the article by Ranzini and Lutz (2017) [15] observed that their participants scored higher in authentic than in deceptive self-presentation. People with low self-esteem tended to construe a more deceptive virtual self as opposed to people with high self-esteem. Dating platform users whose main motivation was to attract potential partners were more likely to misrepresent information about themselves than those more motivated by being seen as truthful.

Ranzini and Lutz (2017) [15] also explored different motivations for using dating platforms. People whose main motivation for using such networks was making new friends created a more authentic virtual self, whereas those who were motivated by self-validation were less authentic in creating their virtual self. On the other hand, those who were looking for casual "hook-ups" were more deceptive, while those who were looking for long-term relationships presented a less deceptive virtual self.

On professional networks, Sievers et al. (2015) [43] reported that people presented themselves authentically in the extraversion, openness, flexibility, and resilience traits. None of the traits explored by the study seemed to be significantly idealized on professional networks. When explicitly instructed to create a profile that represented an idealized version of themselves, professional network users showed a smaller variance in their profiles than when instructed to present themselves accurately. This indicates that although people do not especially enhance the way they portray themselves on professional networks, there seems to be a common understanding on how an ideal profile should be.

## 4. Discussion

In this review, we analyzed the literature about the different ways in which people created their virtual identity in different online environments and how they relate to their personal identity. Although most of the research conducted on the topic distinguishes almost exclusively between accurate and idealized virtual selves [1,3,9,34,36,41], other studies have shown that these are not the only forms of virtual self construal. Refs. [10,11,21] observed the exploration of alternative possible selves that are not necessarily an idealized version of the person's self. This refers to what Taylor (2002) [4] named "identity tourism", the idea that people can experience satisfaction just by taking on a different identity, expanding the self rather than compensating for perceived lacks [4,44]. Another observed type of virtual self is the "negative true self", the expression of negative or inappropriate characteristics that the individual still sees as a part of themselves in order to increase their

self-congruence, according to Hu et al. (2017) [7]. A similar "negative" type of virtual identity was also observed by [8]. People may use the anonymity of online environments to embody personality traits that they cannot express offline since they would be socially unacceptable and cause them to be rejected or even because they would not feel comfortable expressing due to the dissonance it would cause with their offline identity. In this case, the psychological role of virtual identity might be related to the release of repressed parts of the self and increasing self-congruence.

These different forms of virtual self construal seemed to vary depending on the virtual environment. On social media and in specialized environments, virtual identity has tended to be divided between accurate and idealized/enhanced. On the other hand, in more customizable environments such as online video games or avatar-mediated environments, there was a higher tendency to explore alternative or negative aspects of the self. It must be said, however, that this difference could be affected by a bias regarding the objectives of the studies that explored them. Except for a few studies, such as the ones by Fullwood et al. (2016, 2020) and Hu et al. (2020) [10,11,42], research focused on social media has rarely assessed virtual identity construal outside of the "accurate vs. idealized" dichotomy. On the other hand, for video games or other avatar-mediated environments, research has adopted a broader approach to virtual self typologies. Our results indicate that a unified and comprehensive conceptualization of the role of virtual self construal is necessary to better inform future research and clarify the way people are expanding, expressing, compensating, or enhancing their identity through a virtual self.

Another important factor that can impact on how people construe their identity online is the motivation or purpose for using the online environment [6,13–15]. This can be related to the type of environment, since although people can have different reasons for using a specific online platform, each type tends to facilitate certain forms of interaction more than others. People who sought to attract possible partners online tended to present themselves in a more idealized way across different types of virtual environments [13–15], although an exception was found when they were seeking long-term relationships [15]. In this case, their virtual self construal was more similar to their offline identity. When the motivation for using online environments was more playful or entertainment-driven, alternative identities were more likely to be adopted [6], and even negative versions of the self, especially when the users wanted to maintain their anonymity [21,42].

Despite these particularities, the most common pattern of virtual identity construal was that of similarity with the user's actual self [10–12,15,21,43]. However, it must be noted that this pattern was mainly found on social media and in specialized environments, with it being less frequent in avatar-mediated environments and not being particularly common in any of the video game-focused articles. This could be explained partially by the fact that social media and specialized environments encourage their users to present themselves as they are offline, even if they offer the opportunity of self-enhancement and even if most people take that opportunity to some degree. On the other hand, avatar-mediated environments and, to a much higher degree, video games are actively centered on a fictional character created by the user. Additionally, the level of virtual-self customization offered by video games and avatar-mediated environments is much higher than that of social media or specialized environments. Being able to create and customize a virtual avatar, especially a more sophisticated three-dimensional one, offers a lot more options for identity exploration than a platform limited to a collection of profile pictures and a biography. These limitations inherent to the specific virtual environment could also partly explain why virtual identities that deviate from the offline self are rarer on social media platforms, although not completely unheard of, as shown by the phenomenon of Twitter role-players [45].

Nevertheless, we found certain variables regarding their personal identity that can modulate this pattern. Self-esteem, and other similar constructs were among the most influential. In most online environments, the participants tended to create a more idealized virtual self when their self-esteem was lower [3,15,16,32], although this idealization tended

to be more oriented towards reducing their own negative traits more than enhancing the positive ones [1]. On the other hand, when self-esteem was high, people tended to create a virtual self that resembled more how they were offline [9,15]. However, one article provided contradictory evidence. Forest and Wood (2012) [9] reported that people with low self-esteem tended to post more negative content on social media. This could indicate the presence of some exogenous variables that may influence the relationship between self-esteem and virtual self construal. For example, in the study by Forest and Wood [9], their low self-esteem participants reported perceiving social media as a safer space for identity construal than their offline environment, which could be related to their type of virtual self construal.

Self-concept clarity was another relevant factor for virtual self construal, although it was only studied in the context of social media. The less clear a person's notion of their own identity was, the more they tended to explore multiple alternative or enhanced identities on social media [10–12]. However, higher clarity did not always indicate the construal of a virtual self that was similar to the offline person [12]. This seems to indicate that, although people who do not have a clear idea of their identity may tend to explore more with their virtual self, people with a clearer sense of self do not always express their offline identity within virtual environments either. More research should be carried out to explore the cause of this phenomenon and assess how it applies to different virtual environments.

The research included in this review also reports different ways in which virtual self construal could be related to different modalities of personal improvement and psychological well-being. One interesting finding was that, in some cases, ideal traits assigned to a person's virtual identity could eventually carry over into their offline self. This effect was reported in avatar-mediated environments, where people who created, for instance, a more extroverted virtual self also ended up being more extroverted offline [40]. Similar research has also been carried out regarding video games, where this phenomenon is known as the "Proteus effect" [22], and other research has even found that these effects can carry over once the person goes offline, according to both their own experience and their results in evaluation questionnaires [46]. A possible explanation for this could be that, since virtual platforms can be perceived as safer in terms of social consequences, they are good environments for playing and rehearsing different roles, which improves the user's plasticity. Once integrated, a desired feature can make its way to the offline realm. Although more research is needed, especially in other virtual environments such as social media, the capacity for virtual settings to be used for self-improvement has been discussed since the coinage of the term "serious games" by Abt (1970) [47], and it has already been studied to some extent, especially on highly immersive platforms such as virtual reality [48]. Some of the reviewed studies suggested that motivation for using specific virtual environments also affected the benefits of embodying an idealized identity in avatar-mediated environments. While people who used these environments for socialization seemed to benefit more from it, those who were there mainly as a form of evasion from the offline world experienced lesser benefits to their psychological well-being [13].

The exploration of the negative self in online environments has also been found to be related to higher psychological well-being [42]. People can achieve self-congruence within the online environment by expressing those traits they consider inappropriate to show offline, which contributes to their satisfaction and happiness. However, this characteristic of online environments also comes at a risk of harmful self-confirmation bias, since it is possible for undesirable behaviors to be reinforced by like-minded individuals in heavily secluded virtual groups.

As a final observation, something that captured our attention when reviewing the included studies was that people tended to have a distorted notion of how idealized or realistic their virtual self was in online environments. Even when people reported creating virtual identities that were similar to them, third-party raters observed that they had represented themselves in a more enhanced way [41]. This observation is relevant for any research that has been undertaken about this topic. Most of the studies used methodologies

that included very transparent questions about themselves and their virtual self construal, which made them sensible to social desirability. Future research should take into account bias when choosing the methodology, favoring procedures that capture implicit processes and relationships of self construal contextualized in the construing of significant others, such as the repertory grid technique [5,49].

## 5. Limitations and Future Research

Despite the measures taken to reduce bias in all of the critical steps of the review, this study still presents some limitations. First, there could be relevant articles, perhaps in the grey literature, that met the inclusion criteria but were not identified by the search strategy. Even after using the hand-search and snowball strategies, some relevant articles could have gone undetected. Additionally, the number of articles identified within the "specialized online environment" category was very small. This might reflect the fact that research conducted on online dating or professional networks is less focused on personal identity, given the highly specific purpose of such platforms, as opposed to social media or video games. There were also limitations in terms of inter-rater agreement. As mentioned, although the agreement in the filtering process there was good, in the quality assessment process, the kappa index was poor and, therefore, those ratings have to be taken cautiously.

Furthermore, the number of relevant articles that met the inclusion criteria was limited. Along with the diversity of approaches and different types of virtual environments, this prevented the use of more robust methodologies like systematic reviews or meta-analyses. As more research is conducted on this topic, future reviews employing such methods should be carried out to gain a more complete understanding of virtual identity and how it relates to an individual's offline self.

Future research regarding personal and virtual identity should focus on specific online environments unless the objectives of the study require a broader scope. Given its influence on how people create their virtual self online, research should also consider the motivations that lead people to use a specific environment. Regardless of the specific environment and motivation, when measuring virtual self identity, we see the need for measures that tap into implicit processes. In addition, we encourage efforts to go beyond constraining dichotomic approaches (accurate vs. enhanced self). Preferably, researchers should take into account the various psychological functions that virtual self can play: self-expanding, self-expressing, self-compensating, self-acknowledging, self-enhancing, self-improvement, self-releasing, and self-congruence, bearing in mind that more than one can be present at the same time. The results from this review suggest the need to promote novel approaches to assess the processes in creating and using the virtual self in online environments.

**Author Contributions:** Conceptualization, A.G., A.M. and G.F.; methodology, A.G. and A.M.; validation, A.M. and G.F.; formal analysis, A.G.; investigation, A.G. resources, A.G. and G.F.; data curation, A.G.; writing—original draft preparation, A.G.; writing—review and editing, A.M. and G.F.; visualization, A.G.; supervision, A.M. and G.F.; project administration, G.F. All authors have read and agreed to the published version of the manuscript.

**Funding:** GF and AMdC have received funding from the Catalan government (Agaur, Generalitat de Catalunya) as an emergent research group (ref. 2021SGR 00666).

**Institutional Review Board Statement:** Not applicable.

**Informed Consent Statement:** Not applicable.

**Data Availability Statement:** Data sharing not applicable. No new data were created or analyzed in this study. Data sharing is not applicable to this article.

**Conflicts of Interest:** The authors declare no conflict of interest.

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
