# Peer review of "What Is the Psychological Role of the Virtual Self in Online Worlds? A Scoping Review"

_mti, doi:10.3390/mti7120109_

Round 1
Reviewer 1 Report
Comments and Suggestions for Authors
Thank you for this publication in this very interesting and it seems under-researched field. I concluded this publication is acceptable but need significant work (however depending on your background maybe its minor and not major in terms of work needed.
There are two areas that I see work needed:
1) The publication throughout is missing referencing and deeper reflection of the field. Especially in the introduction the use of 'Some Studies' 'Many publications' etc. is overused. I'd like you to specifically list those studies and also the authors. I have no way to see who the 'some' or 'many' relates to. Sometimes you use footnotes which I would not advise for references. In many cases the reference is completely missing such as when you mention work around 'identity tourism'.
2) You seem to completely dismiss the context (application/world) in which the identity/avatar is created. It seems to me that this plays a large role simply because I can only create the avatar based on the confines of the context and moreover I am not aware of any standard in which I can carry it over to other application hence create a 'true' representation of myself. This is even the case in simple chat interfaces, I am confined to whatever the app offers me in terms of visual cues. I also don't fully agree with the conclusion, which also significantly lacks reference backup, that individuals choose avatars close to themselves. I would like to agree but as mentioned I would need to see more references and related studies about this.
Overall, very good work but it needs some significant referencing and refinement. Keep in mind without backing up your arguments with references, they tend to come across shallow or not deeply researched.
Author Response
Thank you very much for your comments and for taking the time to read and review our manuscript, we have revised the manuscript according to your indications (with any changed parts highlighted in blue in the new manuscript) and addressed your comments:
1) The publication throughout is missing referencing and deeper reflection of the field. Especially in the introduction the use of 'Some Studies' 'Many publications' etc. is overused. I'd like you to specifically list those studies and also the authors. I have no way to see who the 'some' or 'many' relates to. In many cases the reference is completely missing such as when you mention work around 'identity tourism'.
Thank you for your comment, we have expanded the referencing and further developed key subjects.
Sometimes you use footnotes which I would not advise for references.
We are unsure as to what "footnotes" would mean in this context. As we understand it, it would be a highlighted term or sentence within the text that is further explained in a designated space below the page, which we did not do in this manuscript. Could you please point out where those footnotes are so we can address them?
2) You seem to completely dismiss the context (application/world) in which the identity/avatar is created. It seems to me that this plays a large role simply because I can only create the avatar based on the confines of the context and moreover I am not aware of any standard in which I can carry it over to other application hence create a 'true' representation of myself. This is even the case in simple chat interfaces, I am confined to whatever the app offers me in terms of visual cues. I also don't fully agree with the conclusion, which also significantly lacks reference backup, that individuals choose avatars close to themselves. I would like to agree but as mentioned I would need to see more references and related studies about this.
Thank you very much for your insight. We completely agree with your point that the practical limitations of a specific platform do indeed constrain the possibilities for virtual self construal. We have included a reflection on the context in which the virtual identity is construed in the manuscript.
Reviewer 2 Report
Comments and Suggestions for Authors
Dear Authors,
Thank you for the opportunity to read the article 'What is the psychological role of the virtual self in online worlds? A scoping review'. The aim of the analyses carried out was, among other things, to explore the current state of knowledge related to the ways in which people construct a virtual identity and what psychological function the virtual self can play in their lives.
The strengths of the manuscript presented for review are the literature cited and the review according to PRISMA-ScR guidelines.
I view the realisation of the objectives undertaken by the authors positively. The only perceived shortcoming is the formatting of the manuscript (alignment, paragraph spacing) and missing full stops in places (e.g. in line 245 there is "et al" and should be "et al.").
Author Response
I view the realisation of the objectives undertaken by the authors positively. The only perceived shortcoming is the formatting of the manuscript (alignment, paragraph spacing) and missing full stops in places (e.g. in line 245 there is "et al" and should be "et al.").
Thank you very much for your comments and for taking the time to read and review our manuscript. we have addressed the formatting issues that we could identify.
Round 2
Reviewer 1 Report
Comments and Suggestions for Authors
Aspects were addressed